# Fabrication and Optical Properties of Transparent P(VDF-TrFE) Ultrathin Films

**DOI:** 10.3390/nano12040588

**Published:** 2022-02-09

**Authors:** Yong Liu, Wei-Guo Liu, Da-Bin Lin, Xiao-Ling Niu, Shun Zhou, Jin Zhang, Shao-Bo Ge, Ye-Chuan Zhu, Xiao Meng, Zhi-Li Chen

**Affiliations:** Thin Film and Optical Manufacturing Technology, Key Laboratory of Ministry of Education, Xi’an Technological University, Xi’an 710032, China; lyt808021@sina.com (Y.L.); niuxl0123@163.com (X.-L.N.); zsemail@126.com (S.Z.); j.zhang@xatu.edu.cn (J.Z.); geshaobo@126.com (S.-B.G.); zyc@xatu.edu.cn (Y.-C.Z.); xiaomengxatu@126.com (X.M.); medichen@163.com (Z.-L.C.)

**Keywords:** PVDF nano films, Langmuir-Blodgett (LB), refractive index, optical constant

## Abstract

The films of vinylidene fluoride and trifluoroethylene (P(VDF-TrFE)) are widely used in piezoelectric tactile sensors, vibration energy harvesters, optical frequency conversion materials and organic photo-voltaic devices because of high electroactive, good optical and nonlinear optical properties, respectively. In this work, the multilayer structured ultrathin films were fabricated by the Langmuir–Blodgett technique, and the thickness per layer can be controlled accurately. It was found that as the collapse pressure of P(VDF-TrFE) (25:75) and the optimal dipping value are 60~70 mN/m and 15 mN/m, respectively, a high-density film can be obtained due to the compression of molecules. The surface topography and optical properties of the LB films were characterized by X-ray diffraction, white light interferometer and variable-angle spectrum ellipsometer. It was observed that the films are transparent in the visible region and IR-band, but show a high absorption in the UV band. Besides, the transmittance of the films ranges from 50% to 85% in the visible region, and it linearly decreases with the number of monolayers. The average thickness of per deposition layer is 2.447 nm, 2.688 nm and 2.072 nm, respectively, under three measurement methods. The calculated refractive index ranged from 1.443 to 1.598 (600~650 nm) by the Cauchy-model.

## 1. Introduction

Two-dimensional (2D) materials are the functional ultrathin crystalline films with strong bonding in the plane and weak bonding between planes (van der Waals), which exhibits plate-like shapes and includes graphene, MXenes, black phosphorous, diatomic hexagonal boron nitride, Perovskites, Metal-Organic Frameworks (MOFs), Covalent-Organic Frameworks (COFs), Polymers and Metals [1]. These 2D layers can be integrated into a monolayer (lateral 2D structure) or a multilayer stack (vertical 2D structure), and possess a number of unique chemical, mechanical, optical and electrical characteristics, which have been intensively studied in the past few years and offer numerous novel applications in the optoelectronic fields [2,3,4,5,6,7,8,9]. The recent progress on the fabrication, characterization and applications of various 2D heterostructures are especially reviewed [10]. Many studies have been focused on nanoscale electronic and optoelectronic devices based on two-dimensional (2D) materials and ferroelectric materials, such as graphene, MoS_2_, WSe_2_ and aluminum (Al)-doped hafnium oxide (HfO_2_) [2]. These studies demonstrate that the artificial 2D materials may provide access to new properties beyond their component 2D atomic crystals and hence, have been widely used in the MEMS devices for their excellent performances.

Among those two-dimensional (2D) materials, piezoelectric polymer films may be one of the most prospective directions. Poly (vinylidene fluoride) (PVDF) and its copolymers with trifluoroethylene P(VDF-TrFE) have been extensively researched for decades because of its chemical resistance, excellent mechanical properties (flexible), electroactive responses, including piezoelectric, pyroelectric and ferroelectric effects [11,12,13,14]. This semicrystalline polymer is the most typical ferroelectric (β phase) material with its zigzag trans conformation [15,16,17,18]. The developed electroactive structures of PVDF have shown potential to be used in a wide range of applications, such as the formation of sensors and actuators, in biomedicine, energy generation and storage [14,19,20]. The state-of-the-art BTO/PVDF composites interlayered by 2-layer BTO exhibit the highest energy storage density, which is 222.6% higher than that of pure PVDF [21]. There are also some studies conducted on the optics properties of PVDF, PVDF-copolymers, PVDF-Blends and PVDF-nanocomposites, for its quite large optical windows (200~1200 nm), good linear and non-linear optical (NLO) properties. The well-known applications of PVDF-nanocomposites are used for optical frequency conversion materials, memory and limiting devices [22,23,24,25,26], which are doped with nanoparticles, such as ZnO [27], CuO [28], ZrO_2_ [29], HfO_2_ [30], reduced graphene oxide (RGO) [31,32], Carbon Nanotubes [33,34], Carbon Quantum Dots [35], cellulose [36], Li_4_Ti_5_O_12_ [37], and so on.

Compared with the traditional preparation methods, such as spin-coating, solution-casting and electrospinning [13,14,19], the LB technology can obtain ultrathin films of PVDF and its copolymers as thin as one monolayer (ML). Thus, this method can produce a highly polar and non-centrosymmetric structure [38,39,40,41,42,43,44,45,46,47], and be regarded as 2D materials [47]. Until recently, many researchers focused on the ferroelectric properties [48,49,50,51,52,53,54,55], including ferroelectric-paraelectric phase transitions by means of optical second harmonic generation (SHG) effect [41,42]. The PVDF LB ultrathin films are fabricated with a thickness of 2.3~2.4 nm per monolayer, which have a complete β phase and show remarkable remanent polarization [43,44]. The large-area PVDF LB films are prepared with a thickness of 1.9 nm and RMS roughness of 0.3 nm by the hot-pressing method, which improved structure uniformity [45]. 

Ferroelectric polymer is always polymorphous, containing amorphous material and crystalline phases, and generally has a broad distribution [12,16,17]. So, those experimental values including surface roughness, thickness, refractive indices (n) and extinction coefficient(k) of P(VDF-TrFE) differ substantially from each other. The process of the deposition is a key factor in the optical and other physical properties. So, it is necessary to study the surface topography, optical constants, and dipping process of the LB films, which can be set as a research starting point for understanding the microscopic mechanism governing the optical behaviors of this film. In this work, the optimal surface pressure was firstly obtained by π-A isotherm experiment. The properties of P(VDF-TrFE) LB films were characterized by X-ray diffraction (XRD), White light interferometer (WLI), Atomic force microscopy (AFM), Fourier transform infrared reflection (FTIR), Variable-angle spectrum ellipsometer (VASE), and Ultraviolet-visible spectrophotometer (UV-Vis Spectra). Finally, the Lambert–Beer’s Law and dispersion model were used to illustrate the optical constants and show different values.

## 2. Materials and Methods

### 2.1. Materials

P(VDF-TrFE) (Mw¼ 300,000, VDF: TrFE mol% = 25:75) used in this work was provided by ARKEMA, PIEZOTECH (Columbus, Paris, France) with a density of 1.78 g/cm^3^. The ITO-glasses were purchased from the GULUO GLASS (Luoyang, Henan, China) and Si wafers were GUI JING ELECTRONIC (Shenzhen, Guangdong, China), both of the thicknesses of the two substrates are 1 mm, and the thickness of ITO layer is 200 nm (PV 20 nm). The N’N-dimethylformamide (DMF, AR, 99.5%) was used as the solution of PVDF, acetone (AR, 99.9%) and ethyl alcohol (AR, 95%) used as cleaning reagent were provided by CHRON CHEMICALS (Chengdu, Sichuan, China).

### 2.2. Films Preparation

Pretreatment of P(VDF-TrFE) solution and substrate: The experiment was completed in the national standard super clean room laboratory (grade 10^5^). P(VDF-TrFE) particles were dissolved in DMF to make solution (0.01 wt%), then put the solution into the water bath to 70 °C and stirred for about 5 h, and placed at room temperature over 24 h. ITO-glasses and Si wafers were selected for the substrate, and the substrates were dipped into acetone and ethyl alcohol for 10~20 min and washed with deionized water after ultrasonic cleaning, then baked for 30 min.

Preparation of the P(VDF-TrFE) micro/nano films: Figure 1 shows the fabrication process of P(VDF-TrFE) nano films. The nano films were performed with a KSV-NIMA system, the solution was spread onto the surface of ultra-pure water for (10~30) mL with a syringe. After the thorough evaporation of the DMF solvent for 10~40 min, the polymer molecules were compressed with a movable barrier until they formed close packed structure. Then π-A curves of isotherm experiment and collapse pressure were obtained by LB software in KSV-NIMA system. The deposition of nano films was assembled by horizontal lifting mode and deposited onto the ITO-substrate at the surface pressure of 15 mN/m [38]. The microfilms were mainly prepared by spin-coating (conventional spin coater, 750 rpm) with the same substrate and solution. The dipping tools were a microliter syringe and a dropper for different concentrations of 0.1 wt% and 0.01 wt% [13].

### 2.3. Characterization

Microstructure and surface morphology were measured with XRD-6000 (Shimadzu Inc., Saitama Prefecture, Tokyo, Japan) (conditions: Cu target, 40 kV, 30 Ma, scanning speed 10°/min), WLI (New View 8300, Zygo Corp., Laurel Brook Road, Middlefield, CT, USA), AFM (MultiMode 8, Bruker Corp., Billerica, MA, USA). Transmittance was carried out using UV-Vis Spectra (U-3501, Hitachi Inc., Chiyoda District, Tokyo, Japan) (spectrum:185 nm~3200 nm, wavelength precision: ± 0.2 nm, near infrared light: ± 0.2 nm, Wavelength resolution: 0.1 nm). FTIR spectra was recorded with 60SXR-FTIR (Nicolet Corp., Madison, WI, USA), wavenumber range: 400 cm^−1^~4000 cm^−1^. Refractive index, thickness and extinction coefficients were obtained by VASE (M-2000 UI, J.A.Woollam Corp., Lincoln, NE, USA) over the wavelength range from 250 nm to 1700 nm, accuracy: 0.2°, repeatability: 0.005°, and angle range 45°~90°.

## 3. Results and Discussions

### 3.1. Deposition Mechanism of Ultrathin Films

As known, the exact surface or collapse pressure is the premise of the dipping. There are few reports about how to obtain and determine the collapse pressure of PVDF or P(VDF-TrFE). Table 1 displays the comparison of the collapse and surface pressure in this work and some published work. In order to obtain the exact collapse pressure, 12 target-values have been set in the π-A experiment for 5, 10, 15, 20, 25, 35, 40, 45, 50, 55, 60 and 65 mN/m, respectively. 

As shown in Figure 2, the P(VDF-TrFE) molecules were pushed closely, and this process can be divided into four stages as the barrier was compressed. In stage I, when the barriers moved from 250 cm to 165 cm, the surface pressure increased slowly from 0.818 mN/m to 5 mN/m. This means that the hydrophobic end—“F” atom of P(VDF-TrFE) molecules begins to stand up and compress from the “gas-liquid” quasi free state to “liquid” condensed phase. With the surface pressure increasing from 5 mN/m to 15 mN/m in stage II, the slope of the curve is markedly elevated, which signifies that a liquid film has formed. When the surface pressure rises quickly from 20 mN/m to 40 mN/m in stage III, polymer molecules completely erect and the stable solid phase films form. Until the barrier moved to the end (25 cm), the curve exhibits a tipping point at which the condensed phase transition changes from III to IV, the slope decreases and the films can begin to be packed or be seen as the monolayer collapsing. Unfortunately, the exact collapsing pressure has not been found. In this isothermal experiment, the collapse pressure of P(VDF-TrFE) can be over (65~70) mN/m, which is close to 60 mN/m [40], much greater than the value of 23 mN/m [38] and 18 mN/m [49]. In practice, a surface pressure cannot be higher than 72.8 mN/m, which is the surface tension of pure water [53].

Figure 2 demonstrates that P(VDF-TrFE) molecules begin to form a monolayer when the surface pressure increases quickly to 5 mN/m, the slope is remarkably increasing. So, the stable monolayer can form on the ultrapure water surface with a collapse pressure at (10~20) mNm^−1^. This can be the main reason that the senior researchers [38,49] have selected “15mN/m” as the optimal value of dipping surface pressure. In addition, from Table 1, it has also displayed that the concentration of P(VDF-TrFE) solution is not necessarily related to the collapse pressure.

### 3.2. Structure and Surface Morphology

As a semicrystalline polymer, the lamella of PVDF or P(VDF-TrFE) bulk materials and thick films (>30 μm) are embedded among the amorphous phase. The LB films are polycrystalline with chains parallel to the horizontal direction of it, and seem to be without lamellar [48,54]. Figure 3a,b show XRD patterns of PVDF and P(VDF-TrFE) LB films. It was found that P(VDF-TrFE) exhibits the peaks at 2θ ≈ 20.5°, 21.44°, 22.8°, 29.44° and the peak intensity represents the sum of diffraction peaks about crystallographic plane (110), (200) of β phase. It also exhibits peaks at 2θ ≈ 17.64°, 18.04°, 26.84°, corresponding to diffraction from (100), (200) of α phase. It can be concluded that: (i) the peak intensity of P(VDF-TrFE) is much stronger than PVDF, which can be attributed to the molecular polarity and rotational barriers of the PVDF molecules at room temperature. It is difficult to obtain the stable β-PVDF at normal temperature and pressure, unless TrFE is added to the original PVDF molecular chains by certain proportion (50~80%). The atomic radius of positive hydrogen is larger than that of negative fluorine atoms. When the VDF and TrFE units are randomly distributed along the molecular chain to form a random copolymer, and the rotational barrier will prevent the chain from forming trans-configuration (β and few γ); (ii) Heating leads to phase transition (α → β). High temperature treatment above the Curie points (135 °C) to 140 °C, which can stimulate a large number of α-PVDF (orthorhombic) molecules to transform into β phase (monoclinic), with strenuous dipole movements [12,16].

The WLI is widely used in the measurement of surface 3D topography of optical films. As shown in Figure 3c,d, there is a clear boundary line between the deposition area and non-deposition, and clear deference in morphology of the P(VDF-TrFE) LB films with different thicknesses (ML). The films became thinner and the roughness rose with ML increasing. The thickness of 10 mL was 20.617 nm, less than that of 100 mL, the surface roughness rose sharply (5.977 nm → 13.4016 nm), which is explained with the same conclusion of AFM. 

The LB films always have a weak binding force with the substrate, the tapping-mode of AFM has been adopted to reduce the force between the tip and the films, and shorten contacting time, which is compared with the contacting-mode. Figure 4 displays that both PVDF and P(VDF-TrFE) films by the LB technique are thinner, than they are with spin-coating. Especially, the monolayer thickness by LB is about 2nm, which is in good agreement with Palto [38]. On the other hand, the thickness of the films by spin-coating is 18.0~48.2 nm. The former is an order of magnitude smaller than the latter, which leads to the change of transmission spectrum as shown in Figure 5. Furthermore, the former was more compact and uniform than the latter. 

It is worth paying more attention to the existence of an obvious crystal boundary among the films by spin-coating, which, due to P(VDF-TrFE) macromolecules had folded multilayers and agglomerated together to form a peak or island structure on the substrate (see Figure 4c,d). The blue oblique lines (see Figure 4a,b) are the place where the film deposited sufficiently (dot A, B, C), which indicates the crystallographic plane (110) with a 45° orientation. It proves that a large number of β-PVDF crystallites have appeared. The preparation method for the films, such as the spin-coating method, mainly depends on the direct adsorption on the surface, and the orientation and arrangement of molecules are difficult to control. However, the molecules of LB films can orient better with the high surface pressure and self-assembly, after being adsorbed on the surface of fluid subphase [56]. All in all, the LB method is better than the spin-coating for preparing an ultrathin film of P(VDF-TrFE).

In Figure 4a, the total thickness of P(VDF-TrFE) by LB with 30 mL and 60 mL are 55.2 nm and 161.0 nm, respectively. However, the monolayer thickness is obviously thicker than the former. The main reason is that the interlayer spacing became wider [57]. Firstly, as the deposition continued, the solution became diluted and the TR (transfer ratio, represents the quantity and the quality of the deposited monolayer on a solid support) decreased [56]. After deposition, the moisture increasing and retention time was not enough to dry sufficiently, and the mixture with the polarity H_2_O molecules strengthened repulsive force intermonolayer, resulting in the LB films thicker. Secondly, after the treatment at high temperature above the Curie point, H_2_O molecules escaped, although the crystalline orientation of polymer monolayer had been enhanced, the relaxation effect existed. At the same time, the gap still exists with H_2_O molecules evaporation, and the interlayer distance had no significant change. On the whole, the thickness of the LB multiplayer increased.

### 3.3. Optical Properties

#### 3.3.1. FTIR Spectroscopy

As shown in Figure 5, the transmittance of P(VDF-TrFE) LB films was obviously the maximum and reached 100%, equal to glass (SiO_2_), from 4000 cm^−^^1^ to 7000 cm^−1^. On the whole, the transmittance of PVDF and P(VDF-TrFE) was high in IR-band or high absorption in UV band. It demonstrates that the PVDF polymer, as an unconjugated polymer, has a maximum transmittance in IR, that was, in the low energy state, even the P(VDF-TrFE) molecules in the ground state (S_0_) may absorb a single photon. But the energy with the absorption from S_0_, was not enough to complete the transition to singlet excited state S_1_ or S_2_ [58]. So, the incident light could pass through easily, resulting in a high transmittance. 

In addition, it has been found that the more dilute the PVDF solution (0.01 ˂ 0.1%) was, the more transparent the films were (graph 1 and graph 2). This can be attributed to the high concentration, which leads to P(VDF-TrFE) molecules fully spreading on the surface of the subphase, and the films were compact to decrease the transmittance. Furthermore, the results showed that the films by different preparation methods have a different impact on the transmittance (graph 2 and graph 3). The films prepared by the LB method were much thinner than by dropper and microliter syringe, so the transmittance of the former was higher than the latter at the same wavelength. Lastly, a fairly qualitative but nonetheless significant observation was that the transmittance of P(VDF-TrFE) LB films was higher than that of PVDF. The reason may be that, for P(VDF-TrFE) molecules, VDF and TrFE units are randomly distributed along the molecular chain to form a random copolymer and cocrystallize into a single crystalline phase analogous to β-PVDF [57]. The crystallinity and orientation of polymer increased, with “n” decreasing, and “T” increasing correspondingly [59].

#### 3.3.2. UV-Vis Spectroscopy

As shown in the UV-Vis spectroscopy of the P(VDF-TrFE) LB films (Figure 6a), it shows that there are three points of high transmittance points (~463 nm, ~611 nm, ~858 nm) and three low transmission points (~424 nm, ~530 nm, ~664 nm) in the visible region. The transmittance reduces linearly with the number of MLs increasing, which proves the regularity of LB films we prepared. 

The mechanism of this phenomenon can be explained with the Lambert-Beer’s Law [60]:(1)A=εCL,
where *A*, *C* and *L* are the absorbance, concentration, and optical length of media or solution, respectively. For thin films, *C* and *ε* are constants, so  A∝L, and T∝1/. Thus, the transmittance of 30 mL is larger than that of 100 mL and 200 mL at the same wavelength. Although the LB films were obtained by multiple times and the short interval among each deposition, there is not enough time for P(VDF-TrFE) or PVDF molecules to solidify, owing to the weak Van der Waals force among the films and the substrates. So, they can fully stretch and intersperse, and the films could be regarded as a monolayer (see Figure 6b). Because of the anisotropy of semicrystalline polymer, the refractive index of P(VDF-TrFE) films shows a significant change with the inhomogeneity of the films. There exists absorption in the amorphous orientation, and the refractive index (n) must change with the dielectric constant frequency of incident light wavelength (Formula (3)), which is consistent with band theory [61]:(2)$$\varepsilon =\varepsilon_{1}\left(\lambda \right)\;+\;i\varepsilon_{2}\left(\lambda \right)$$
(3)$$\alpha =4\pi Κ/\lambda =A{\left(h\upsilon -{Ε}_{g}\right)}^{m}$$

#### 3.3.3. Optical Constants

Refractive index (n) and extinction coefficient (k) are the main optical parameters to evaluate the physical properties of anisotropic optical films. The ellipsometry shows easy operation, simple data processing, nondestructive and high accuracy, promoting an appropriate method to measure the n, k and thickness of ultrathin films [61,62].

As known, no matter what kind of optical films, chromatic dispersion inevitably appears. PVDF or P(VDF-TrFE) is the typical transparent polymer with no or low absorption, like PE, PMMA, PC [56,57,58,59,60,63]. It can be wise to use the Cauchy relations to simulate and calculate for PVDF or P(VDF-TrFE), for less parameters and simple model [38].

In Figure 7, all the curves of n and k are basically in accordance with dispersion law, the value of n or k decreases with the λ increasing, which is according with the Cauchy relations.
(4)nλ = An + Bnλ2+Cnλ4+⋯
(5)kλ = Ak+Bkλ2+Ckλ4+⋯

The values of n are all over 1.40, which increase with thickness in the visible region (380~800 nm), indicating that PVDF and P(VDF-TrFE) films are the typical antireflective film, which confirms the previous results of the UV-Vis spectra. Every kind of film has a small amount of absorption, the k of 60 mL, 160 mL, 200 mL, is constant value, meanwhile, the *k* of 200 mL is a variation by dispersion-model. The absorption of 60 mL, 160 mL and 200 mL are strong (~0.1382, ~0.1601). It is worth noting that the maximum of n belongs to 30 mL, whose curve has a tendency of anti-dispersion when λ > 675 nm, such as 60 mL and 160 mL. The main optical parameters of P(VDF-TrFE) LB films could be obtained associated with the calculation (Table 2).

Figure 8 shows the mean thickness of P(VDF-TrFE) monolayer are 2.072 nm by AFM, 2.447 nm by VASE and 2.688 nm by WLI, which are not significantly related to the refractive index (*n*) and extinction coefficient (*k*). In addition, the measurement standard error of VASE is the maximum, which was attributed to the scattering of inhomogeneity ultrathin films.

In this work, the experimental values of n are larger than calculated values by first-principles theory [23,51,54], as well as larger than the experimental value [22,24,25,49,50,64]. The only reason can be the preparation technology of P(VDF-TrFE) films. It demonstrates that refractive index of LB films (nanometer scale) is higher than by spin-coating (micron scale), which agrees with the conclusion of FTIR spectra, AFM and WLI. The large k is attributed to the photon absorption and scattering, which is caused by pores and defects of the films [65,66,67]. As shown in Figure 9, the films of PVDF or P(VDF-TrFE) exhibit almost the same optical properties within the same scope, whether its thickness is in the nano scale or the micro scale.

## 4. Conclusions

P(VDF-TrFE) transparent ultrathin films were fabricated by the LB technique. The range of collapse pressure is from 65 mN/m to 70 mN/m, and the dipping value of the LB films is 15 mN/m, which could be explained with the theory of polymer molecular condensing and stacking at different stages. The LB films show transmittance in the IR-band and high absorption in the UV band because of the energy band and optical absorption of P(VDF-TrFE) intrinsic characters. The transmittance of the films over 50~85% in the visible region, which decreased linearly with the number of MLs. Meanwhile, both the thickness of the monolayer and the TR decrease with the concentration of solution. The refractive index is between 1.443~1.598 (600~650 nm). Therefore, these results imply the P(VDF-TrFE) optical LB films have a promising application in the transparent MEMS devices in the future.

## Figures and Tables

**Figure 1 nanomaterials-12-00588-f001:**
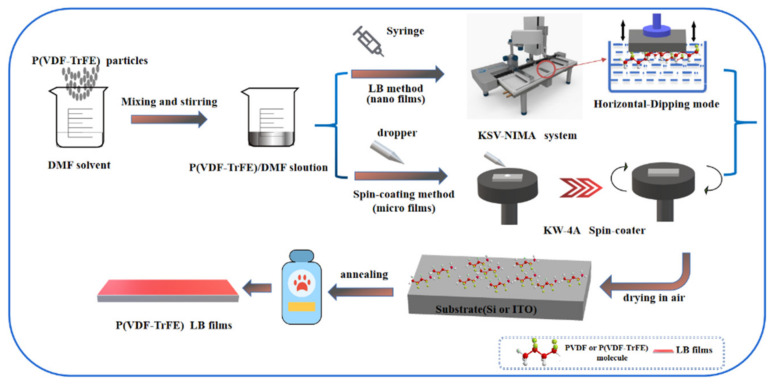
Schematic of preparation of P(VDF-TrFE) ultrathin films.

**Figure 2 nanomaterials-12-00588-f002:**
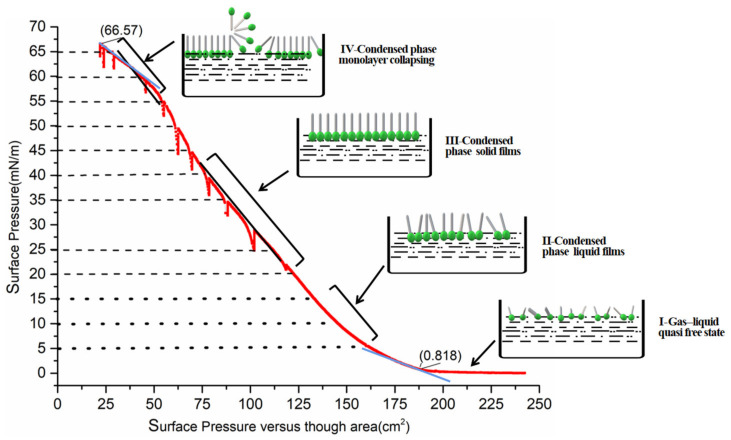
π–A isotherm and the LB films formation of P(VDF-TrFE) Solution (0.01%) with ultra-pure water subphase.

**Figure 3 nanomaterials-12-00588-f003:**
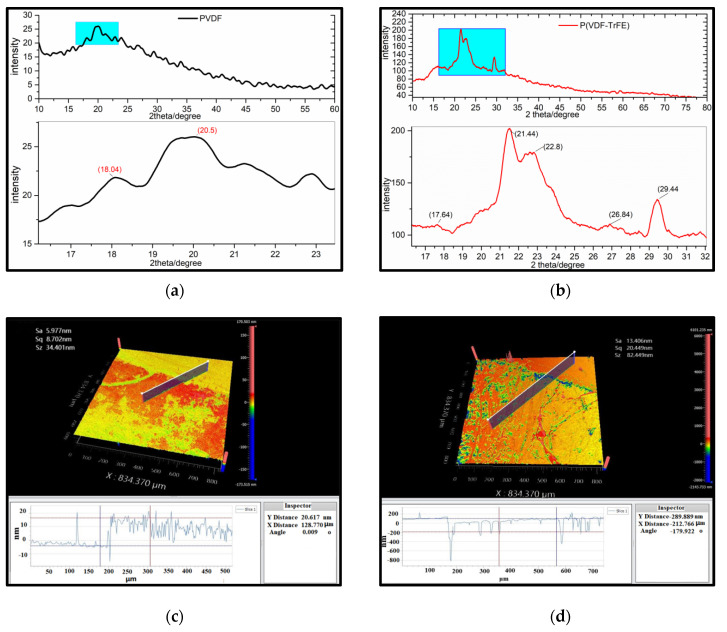
X-ray pattern of (**a**) P(VDF-TrFE) and (**b**) PVDF; WLI of P(VDF-TrFE) by LB with different ML: (**c**) 10 mL, (**d**) 100 mL.

**Figure 4 nanomaterials-12-00588-f004:**
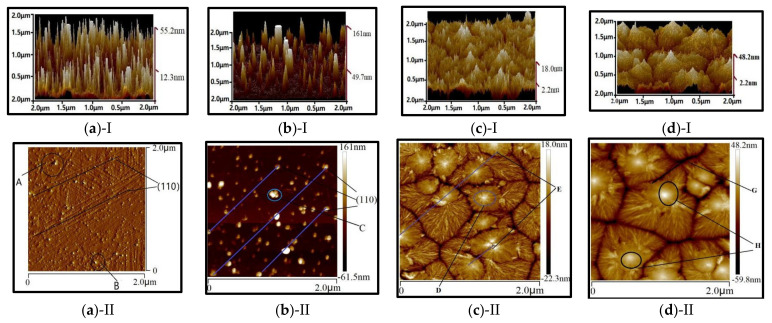
AFM of P(VDF-TrFE) by LB and spin-coating with different ML and different concentration: (**a**) 30 mL; (**b**) 60 mL. (**c**) P(VDF-TrFE) (0.1 wt%) and (**d**) P(VDF-TrFE) (0.01 wt%) by spin-coating.

**Figure 5 nanomaterials-12-00588-f005:**
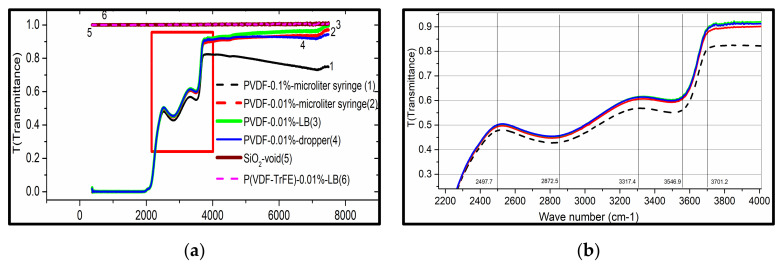
(**a**) FTIR spectra of PVDF and P(VDF-TrFE) films by different method; (**b**) the partial enlarged view of (**a**).

**Figure 6 nanomaterials-12-00588-f006:**
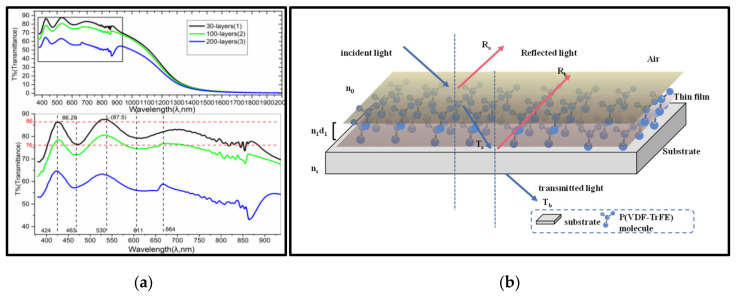
(**a**) UV-Vis spectra of P(VDF-TrFE) LB multiplayer films with different ML; (**b**) The optical path schematic of P(VDF-TrFE) monolayer film.

**Figure 7 nanomaterials-12-00588-f007:**
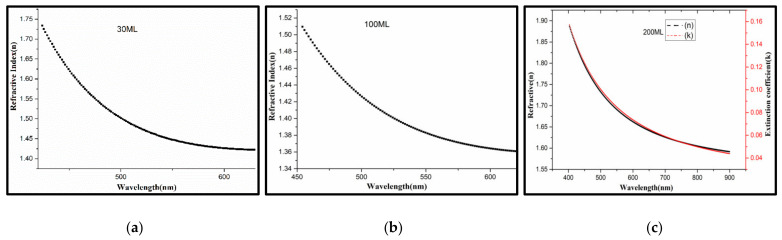
Variation of n, k of P(VDF-TrFE) with incident-light wavelength and ML fitted by Cauchy-model: (**a**) 30ML; (**b**) 100ML; (**c**) 200ML.

**Figure 8 nanomaterials-12-00588-f008:**
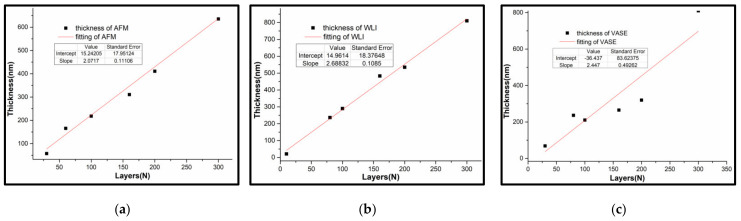
The film thickness vs. the number of MLs: (**a**) WLI; (**b**) AFM; (**c**) VASE.

**Figure 9 nanomaterials-12-00588-f009:**
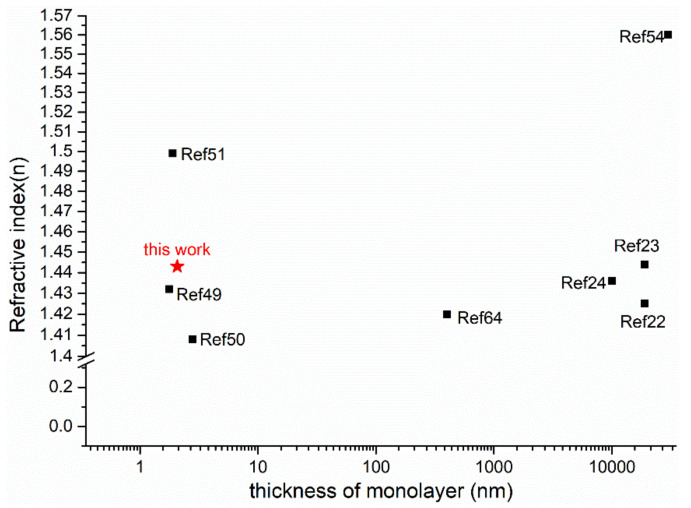
Comparison of the optical parameters in PVDF nano and microfilms.

**Table 1 nanomaterials-12-00588-t001:** Comparison of the preparation conditions and parameters for PVDF ultrathin films.

Collapse Pressure (mN/m)	Surface Pressure for Dipping (mN/m)	Solvent	Solution Concentration	Dipping Mode	Reference
60–70	15	DMF	0.01 wt%P(VDF-TrFE)	Horizontal	This article
6	15	DMSO	0.01 wt%P(VDF-TrFE)	Horizontal	38
5–20	5	DMSO	0.01 wt%P(VDF-TrFE)	Horizontal	49
60	40	DMF	0.1 wt%-PVDF	Y	40
--	5	DMF	P(VDF-TrFE-CFE) 0.01wt%	Horizontal	50

**Table 2 nanomaterials-12-00588-t002:** The Parameters of P (VDF-TrFE) ultrathin films with Cauchy-model.

Number of ML	Fitting Parameters					
	*A_n_*	*B_n_*	*C_n_*	*A_k_*	*B_k_*	*C_k_*
30	1.5978	0.1476	0.0309	0.1178	0	0
60	1.4736	−0.1022	0.0226	0.1382	0	0
160	1.5499	0.0286	0.0043	0.1601	0.7523	0
200	1.443	0.0074	0.0092	0.0124	0	0

## Data Availability

The data presented in this study are available upon request from the corresponding author.

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
