# Peer review of "Fabrication and Optical Properties of Transparent P(VDF-TrFE) Ultrathin Films"

_nanomaterials, 2022, doi:10.3390/nano12040588_

Round 1

Reviewer 1 Report

The manuscript describes LB technique to prepare P(VDF-TrFE) nanoscale films which is helpful to understand the technique. However, the flow of the manuscript is not present as extensive English fluency is needed. Also, the manuscript contains text stating "errors" at the place of references which make it difficult to follow the science with the literature. Please correct these issues before another review can be done.

Author Response

Dear Professor, Thank you very much for your review. Please see the attachment.

Reviewer 2 Report

The manuscript “Process and optical properties of P(VDF-TrFE ) nano films” presents the films of vinylidene fluoride and trifluoroethylene prepared by Langmuir-Blodgett technique, their surface topography, and optical properties. The goals and motivation of such a study are relatively clear. The manuscript is well-structured and presented.

The paper can be recommended to Nanomaterials.

Additional comments that need to be addressed:

  • The manuscript is slightly sloppy; compound numbers are not bold in many places, no spaces in lots of places, the punctuation marks (commas, dots) are missed in some places etc., English language and style (grammar) must be carefully revised throughout the whole manuscript. A significant number of typographical/grammatical issues need to be dealt with, for instance:
  • The abstract: ‘The films of vinylidene fluoride and trifl-uoroethylene (P(VDF-TrFE))(25:75) could be widely used for optoelectronic devices. Herein,a physical method etching method for the multilayer structure of films were prepared by Langmuir-Blodgett technique,and shown the accuracy of monolayer can be controlled.It is founded that the collapse pressure of P(VDF-TrFE) may be over(65~70)mN/m,and the optimal dipping value may be setted to 15mN/m ,for the molecules’s compressment closely.The surface topography and optical properties of the LB films were investigated by X-ray diffraction,White light interferometer, variable-angle spectrum ellipsometer and other mesurements. The results showed that the films are transparent in visible region and IR- band,but high absorption in UV band,and the transmittance over 50%~85%, close to glass (SiO2) (visible region),which decreased linearly with the number of monolayer.The thickness averaged 2.447/2.688/2.072nm per deposition layer, the refractive index ranged from 1.443 to 1.598( 20 600nm~650nm)by Cauchy-model.’ should be ‘The films of vinylidene fluoride and trifluoroethylene (P(VDF-TrFE)) (25:75) could be widely used for optoelectronic devices. Herein, a physical etching method for the multilayer structure of films was prepared by the Langmuir-Blodgett technique and shown the accuracy of the monolayer can be controlled. It is found that the collapse pressure of P(VDF-TrFE) may be over(65~70) mN/m, and the optimal dipping value may be settled to 15 mN/m, for the molecules’ compressment closely. The LB films' surface topography and optical properties were investigated by X-ray diffraction, White light interferometer, variable-angle spectrum ellipsometer, and other measurements. The results showed that the films are transparent in visible region and IR- band, but high absorption in the UV band, and the transmittance over 50%~85%, close to glass (SiO2) (visible region), which decreased linearly with the number of the monolayer. The thickness averaged 2.447/2.688/2.072 nm per deposition layer, and the refractive index ranged from 1.443 to 1.598( 20 600 nm~650 nm) by Cauchy-model.’
  • The sentence: ‘But the energy with the absorbtion from S0( 3.b), was not 237 enough to complete the transition to singlet excited state S1 or S2[55].’ should be ‘But the energy with the absorption from S0 (Figure. 3. b), was not 237 enough to complete the transition to singlet excited state S1 or S2 [55].’
  • The sentence: ‘The transmittance decreased linearly with the number of ML increasing,which proved the regularity of LB films we prepared63.’ should be ‘The transmittance decreased linearly with the number of ML increasing, which proved the regularity of LB films we prepared [63].’
  • The sentence: ‘As shown in the UV-Vis Spectroscopy of the P(VDF-TrFE) LB films(figure.5.c-d), It shows:3 absorption bands(~463nm,~611nm,~858nm) and transmission bands(~256 424nm,~530nm,~665nm) in visible region.’ should be ‘As shown in the UV-Vis spectroscopy of the P(VDF-TrFE) LB films (figure.5 c-d), it shows 3 absorption bands (~463,~611, and ~858 nm) and transmission bands (~ 424,~530,~665nm) in the visible region.
  • Please discuss deeper the figure 5 c.
  • The authors claimed: ‘As shown in the UV-Vis Spectroscopy of the P(VDF-TrFE) LB films(figure.5.c-d), It shows: 3 absorption bands(~463nm,~611nm,~858nm) and transmission bands(~424nm,~530nm,~665nm) in visible region.’ Please explain the difference between absorption and transmission bands. It would be nice to present the absorption spectra alongside transmission spectra, if the authors discuss the absorption bands. What about the bands at about 800 830, 870 nm? Why is there such a big difference in transmittance between 100-layer and 200-layer film? Please discuss and explain this.   

Author Response

Dear ProfessorThank you very much for your review. We apologize for the errors of my manuscript. We have now worked for a long time to correct the sentences and sections of my article. Please see the attachment.

Reviewer 3 Report

Dear Authors, 

This study explores the structural and optical properties of ultrathin vinylidene fluoride and trifl-uoroethylene (P(VDF-TrFE) films prepared by Langmuir-Blodgett method. The films were characterized by different techniques and the outcomes highlighted good transparency in the visible region. In addition, the impact of monolayer number is evident on the optical properties of the samples such as their refractive index. Overall, the work is well-organized with including sufficient investigations and its consistence with the general scope of the journal. However, modification and revision of the manuscript is necessary before accepting the publication. Below detailed comments are given:

1-  A better choice of words and phrasing can be very helpful. For example, the title is not as concise and informative as it should/could be. An eye-catching title can attract considerable attention. The authors should pay further attention to such critical matters.

2- There are some grammatical, punctuation and typo errors in the manuscript with lack of academic style writing. Therefore, the English language need to be further polished.

3- Page 4 (line 150 and 151), when comparing your outcomes to the other works you should state their values to help the reader better grasp the message “the collapse pressure of 150 P(VDF-TrFE) may be over (65~70)mN/mwhich is close to Yadong Jiang[39],which is much 151 greater than the value of S. Palto[37]and Mengjun Bai[48]”

4- There are a clear mix up in the referencing, for example in page 8 line (258) the reference suddenly jumped to 63 with different referencing style “  LB films we prepared63”. Then it moves back to reference 59 in page 9 line 268 without existing reference 58. Thus, the referencing and citation should be very carefully revised and unified.

5- All the used equations should be cited.

6- Correct the subheading 3.3.3 Optical Contants  to “ Optical constants”. Many other similar typo mistakes are existing in the manuscript.

7- It is known that ellipsometry is an accurate technique for studying optical properties of thin samples such as refractive index. Moreover, this technique can be used to determine the thickness of the samples accurately. Thus, authors are advised to enrich the manuscript by comparing the refractive index achieved from ellipsometery to those calculated from the UV-visible Spectroscopy and discuss the outcomes.

8- Does this approach can be employed to create anti-reflection coating or gradient refractive index layers for energy device applications like DSSC on the industrial level? 

9- The referencing style should be unified and follow the journal format.

Author Response

Dear ProfessorThank you very much for your review. We apologize for the poor language of my manuscript. We have now worked for a long time to correct the sentences and sections of my article. We really hope that the sections and language level have been substantially improved. Please see the attachment.

Reviewer 4 Report

In the manuscript entitled “Process and optical properties of P(VDF-TrFE ) nano films”, authors Liu Yong, Liu Wei-guo, Lin Dabin, Niu Xiao-ling, ZHOU Shun , Zhang Jin , Shaobo Ge, ZhuYeChuan, describe the obtaining of P(VDF-TrFE)) using Langmuir-Blodget procedure.

Due to misspellings and non-concordances between the text and figures, the manuscript is hardly readable. Therefore, I suggest major revision based on the following aspects.

Minor revisions:

  1. Recheck the whole text and correct it.
  2. References in text. This should be something connected with the formatting of pdf file.
  3. Incomplete figures. There are no concordances between the text and figures (differences between the text and figure caption).
  4. The refractive indexes are noted with “n” in figures and “N” in the text. Therefore, I suggest inverting the number of polymers with “N” and letting the refractive index with “n”.
  5. Move figure 5d to the next page.

Major revisions:

  1. From the beginning is better to make a clear difference between the LB and spin-coating deposition, both in the text and figures.
  2. Use the minima and maxima interference points instead of absorptions and transmissions bands in the UV-viz measurements.
  3. In figure 5d, the substrate is ITO or ITO-on glass substrate? Because in the next calculation there are 4 layers (air, PVDF, ITO, glass)
  4. Connected with that, Ns-is ITO refractive index or ITO on glass? Please revise.
  5. Please, explain what is π-A isotherm.
  6. In XRD figure 3 a), the first peak seems to be over 180 instead of 17.40. Please revise.
  7. The WLI procedure is not well explained. Please revise.
  8. Explain what transfer ratio TR is.
  9. The transmittance of P(VDF-TrFE) LB films reached to 100%, 231 equivale to glass (SiO2), from 4000 cm-1 to 7500cm-1 not from 500 cm-1 and is over 50% only between 2500 cm-1 to 4000 cm-1.
  10. Explain better the figure 5c) and the concordance with 5 b) and c).
  11. Figure 6 c) should be corrected. Which curve is for the extinction coefficient k?
  12. Avoid using non-sense phrases like “ ..55.2 nm, which is thinner than that of …161 nm.
  13. The theory should be revised based on the approximation of only three layers.

Author Response

(The authors gave the same response as above.)

Round 2

Reviewer 1 Report

The authors answered my comments and made changes to the manuscript accordingly. I suggest to publish the article.

Author Response

Dear Professor, thank you very much for your review and comments.Kind regards!

Reviewer 2 Report

The authors provided reliable and comprehensive answers. I think that this revised manuscript is suitable for publication without further revision.

Author Response

Dear Professor, thank you very much for your review and comments. Best regards!

Reviewer 4 Report

In the manuscript entitled “Process and optical properties of P(VDF-TrFE ) nano films”, authors Liu Yong, Liu Wei-guo, Lin Dabin, Niu Xiao-ling, ZHOU Shun , Zhang Jin , Shaobo Ge, ZhuYeChuan, describe the obtaining of P(VDF-TrFE)) using Langmuir-Blodget procedure.

The manuscript was improved but there are some questions and problems:

Minor revisions:

  1. References in text. Still some of them are wrong formatted.

Major revisions:

  1. From the beginning is better to make a clear difference between the LB and spin-coating deposition, both in the text and figures. This fact was suggested to re-order somehow the figures to separate LB by spin-coating technology. For example, figure 4 a) and b) it is supposed to be LB while c) and d) as spin-coating.

  1. In figure 5d, the substrate is ITO or ITO-on glass substrate? Because in the next calculation there are 4 layers (air, PVDF, ITO, glass)
  2. Connected with that, Ns-is ITO refractive index or ITO on glass? Please revise.

If the authors consider ITO on glass as one layer, the mean value of the refractive index is 1.75, but is not accounted the individual reflectance and transmittance of ITO and glass. As a consequence, the transmittances through PVDF and ITO-glass are the same.

  1. The WLI procedure is not well explained. Please revise.

I suggested to explain in details this technique.

  1. The theory should be revised based on the approximation of only three layers.

This is because the transmittances through the whole sample decreases with the number of layers, but only the front face reflectance increase. Meanwhile, the calculated refractive index increases with the number of layers while the measured one by ellipsometry is completely different.

  1. While in the table 2 the refractive index increases with the number of layers, in the figure 7, firstly decrease between 80 ML and 100 ML and then decrease for 200 ML. By the way 80 ML was not mentioned. Probably is 30 ML. Please revise.

Author Response

Dear ProfessorThank you very much for your review and comments. We have corrected the references errors and tried our best to answer the questions. Please see the attachment.

Round 3

Reviewer 4 Report

In this case, delete the theory because it is not proper.

Author Response

Dear Professor, thank you again for your review, and we will happy to edit our manuscript further based on your advice.
